# Blood-Based Biomarker Analysis for Predicting Efficacy of Chemoradiotherapy and Durvalumab in Patients with Unresectable Stage III Non-Small Cell Lung Cancer

**DOI:** 10.3390/cancers15041151

**Published:** 2023-02-10

**Authors:** Cheol-Kyu Park, Sung-Woo Lee, Hyun-Ju Cho, Hyung-Joo Oh, Young-Chul Kim, Yong-Hyub Kim, Sung-Ja Ahn, Jae-Ho Cho, In-Jae Oh

**Affiliations:** 1Department of Internal Medicine, Chonnam National University Medical School, Chonnam National University Hwasun Hospital, Jeollanamdo 58128, Republic of Korea; 2Medical Research Center for Combinatorial Tumor Immunotherapy, Chonnam National University Medical School, Jeollanamdo 58128, Republic of Korea; 3Department of Microbiology and Immunology, Chonnam National University Medical School, Jeollanamdo 58128, Republic of Korea; 4Department of Radiation Oncology, Chonnam National University Medical School, Chonnam National University Hwasun Hospital, Jeollanamdo 58128, Republic of Korea; 5Immunotherapy Innovation Center, Chonnam National University Medical School, Jeollanamdo 58128, Republic of Korea

**Keywords:** circulating tumor cells, platelets, biomarkers, concurrent chemoradiotherapy, durvalumab, non-small cell lung cancer

## Abstract

**Simple Summary:**

This prospective exploratory study suggested the feasibility of circulating tumor cells (CTCs) and peripheral blood cells (PBCs), especially platelets, as predictive biomarkers in patients with unresectable stage III non-small cell lung cancer (NSCLC) who received concurrent chemoradiotherapy (CCRT) and durvalumab. This comprehensive analysis of CTC and PBC counts could provide valuable insights into a biomarker-driven strategy for further management after the completion of CCRT, and could aid in the risk stratification of patients with unresectable stage III NSCLC who are eligible for durvalumab during treatment.

**Abstract:**

We recruited 50 patients with unresectable stage III NSCLC who received CCRT between March 2020 and March 2021. Durvalumab consolidation (DC) was administered to patients (*n* = 23) without progression after CCRT and programmed death-ligand 1 (PD-L1) ≥ 1%. Blood samples were collected before (C0) and after CCRT (C1) to calculate PBC counts and analyze CTCs. CTCs, isolated by the CD-PRIME^TM^ system, exhibited EpCAM/CK+/CD45− phenotype in BioViewCCBS^TM^. At median follow-up of 27.4 months, patients with residual CTC clusters at C1 had worse median PFS than those without a detectable CTC cluster (11.0 vs. 27.8 months, *p* = 0.032), and this trend was noted only in the DC group (*p* = 0.034). Patients with high platelets at C1 (PLT^hi^, >252 × 10^3^/µL) had worse median PFS than those with low platelets (PLT^lo^) (5.9 vs. 17.1 months, *p* < 0.001). In multivariable analysis, PLT^hi^ and residual CTC clusters at C1 were independent risk factors for PFS, and DC group with PLT^hi^ and residual CTC clusters at C1 showed the worst median PFS (2.6 months, HR 45.16, *p* = 0.001), even worse than that of the CCRT alone group with PLT^hi^ (5.9 months, HR 15.39, *p* = 0.001). The comprehensive analysis of CTCs and PBCs before and after CCRT revealed that the clearance of CTC clusters and platelet counts at C1 might be potential biomarkers for predicting survival.

## 1. Introduction

Locally advanced stage III non-small cell lung cancer (NSCLC) could be potentially resectable or unresectable at diagnosis. In patients with unresectable stage III NSCLC with good performance, the historical standard of care for the last few decades was concurrent chemoradiotherapy (CCRT), which consists of radical radiation therapy (RT) and platinum-based doublet chemotherapy [1]. Recently, the phase 3 PACIFIC trial demonstrated the durable benefits of durvalumab, an inhibitor of programmed death-ligand 1 (PD-L1), on overall survival (OS) of patients with unresectable stage III NSCLC [2,3]. After completion of CCRT, durvalumab consolidation (DC) improved the 5-year OS by a hazard ratio (HR) of 0.72 (95% confidence interval [CI]: 0.59–0.89) compared to placebo [3]. The administration of DC for up to 12 months has been the new standard of care in patients with unresectable stage III NSCLC who received CCRT without disease progression during the treatment [1].

In the PACIFIC trial, DC improved the PFS and OS across most subgroups [3,4]. On the other hand, excluding several demographic factors, subgroups with positive driver mutations, such as epidermal growth factor receptor (EGFR) or anaplastic lymphoma kinase (ALK), and PD-L1 tumor proportional score (TPS) < 1% had unfavorable survival after DC, suggesting that those two factors could be tumor-related biomarkers for DC. However, patients harboring driver mutations were a small population, and the PACIFIC trial did not select patients based on PD-L1 expression [2]. In addition, PD-L1 testing was not performed in over 1/3 of the patients included in the trial, and it was performed before CCRT. Re-biopsy after CCRT was usually unfeasible, and the role of PD-L1 as a biomarker for DC could be discounted. Indeed, less than half of patients completed the 12-month course of DC without disease progression in the PACIFIC trial (49%) [3] and real-world studies (43–47%) [5,6]. Considering the efficacy of DC, its high cost, and the long-term exposure to adverse events, reliable biomarkers for DC are necessary to predict the survival benefits.

In recent studies, blood-based biomarkers reflecting the tumor burden and blood microenvironment of the tumor have been drawing increasing attention due to their advantages of liquid biopsy. Pretreatment levels or dynamic changes in peripheral blood cell (PBC) counts, circulating tumor cells (CTCs), circulating cell-free DNA (cfDNA), blood tumor mutational burden, soluble PD-1/PD-L1, and other circulating immune cells or cytokines have been studied as potential biomarkers to predict response to immune checkpoint inhibitors (ICIs) [7,8]. In addition, the role of CTC or circulating tumor DNA (ctDNA) for the assessment of minimal/molecular residual disease (MRD) has been investigated to measure the remnant tumor burden and predict the recurrence and prognosis after curative intent treatment [9,10,11]. However, the validation of these blood biomarkers and application to stage III NSCLC may be limited due to the complex clinical presentation and treatment decision of stage III NSCLC, available robust technologies, the lack of standardized sample preparation methods, the timing of sample collection, and types of liquid samples. Based on these unmet needs, this study aimed at investigating the feasibility of using blood-based biomarkers, such as CTCs and PBCs, in predicting the efficacy of CCRT and DC in locally advanced unresectable stage III NSCLC.

## 2. Materials and Methods

### 2.1. Patients, Treatment and Assessment

This study is a prospective, exploratory, observational trial involving the collection of blood samples from patients with unresectable stage III NSCLC. We enrolled patients with histologically or cytologically confirmed NSCLC who received CCRT at Chonnam National University Hwasun Hospital between March 2020 and March 2021. Patients who completed planned CCRT and had not progressed following the treatment were administered DC. The completion of CCRT was defined as the inclusion criteria of the PACIFIC trial [2], that is, patients had received platinum-based chemotherapy concurrently with RT at a total dose of at least 60 Gy (54 to 66 Gy). Patients who underwent radical surgery before CCRT were excluded.

The first follow-up chest CT scan after CCRT was performed within six weeks after the last date of RT and was repeated every 8 to 12 weeks after that. In the DC group, after the completion of CCRT, durvalumab (10 mg/kg) was administered intravenously for 60 min every two weeks and continued for a maximum duration of 12 months until confirmed progression, initiation of alternative cancer therapy, unacceptable toxicity, or the occurrence of other reasons to discontinue the drug. A follow-up chest CT scan was performed after every four to six cycles of durvalumab. The clinical response to the treatment was defined by the Response Evaluation Criteria in Solid Tumors (RECIST) version 1.1 [12].

### 2.2. Blood Sample Collection and Procession

The study flowchart is shown in Figure 1. Complete blood cells were analyzed to record the number of PBCs, including white blood cells, absolute neutrophil count (ANC), absolute lymphocyte count (ALC), platelet count, neutrophil-to-lymphocyte ratio (NLR), and platelet-to-lymphocyte ratio (PLR), before the first cycle of CCRT (C0). After that, blood tests were performed in both groups as follows: in the DC group, at the initiation of the first cycle (C1) and at each cycle (C2–3–5) of durvalumab, and in the CCRT alone group, at the first follow-up CT scan after CCRT (C1) and at the second follow-up CT scan (C5). Additional whole blood samples (20 mL) were prospectively collected along in K2-EDTA tubes (BD, East Rutherford, NJ, USA) to analyze CTCs and immune cells at the same time points of blood tests (C0 to C5) in both groups, respectively. In the DC group, the C5 samples were collected between the date of the second follow-up CT scan (after the fourth cycle) and the start of the fifth cycle. Blood samplings were performed between the second and fifth cycles (C2–3–5) in available patients. If the DC was terminated before C5, blood samples were collected at the end of treatment (EOT).

Whole blood samples were centrifuged to obtain plasma, platelets, and peripheral blood mononuclear cells (PBMCs). Plasma and PBMC samples were stored in the current institute until the samples were transported to central laboratories (Clinomics, Daejeon, Republic of Korea) for the extraction of cfDNA and enrichment of CTCs, respectively. Whole blood samples were also collected from healthy donors to obtain PBMCs and stored. Details of the isolation of plasma, platelets, and PBMCs and the preparation and quantification of cfDNA are described in Appendix B.

### 2.3. Enrichment, Isolation and Identification of CTC

The CD-PRIME^TM^ system (Clinomics Inc., Ulsan, Republic of Korea), comprising CD-FAST^TM^ solo (disc) and CD-OPR-1000^TM^ (disc operating machine), was used for CTC enrichment [13,14,15,16]. This system had been applied in the authors’ previous study on blood-based biomarkers for ICI in stage IV NSCLC patients. Details of CTC isolation and immunofluorescence (IF) staining procedures were published in the last report of the authors’ study [8]. To visualize CTCs on the filter, fluorescent images were captured using a fluorescence microscope (Bioview, Rehovot, Israel) at 40× magnification. The CTCs were identified, and the number of CTCs was counted according to the following criteria: phenotype, epithelial cell adhesion molecule (EpCAM)/cytokeratin (CK)+, CD45–, and 4,6-diamidino-2-phenylindole (DAPI)+; and diameter, >8 μm (Figure 2).

### 2.4. Detection and Quantification of PD-L1 + Platelets in PBMC

To access PD-L1 expression in platelets, PBMCs from NSCLC patients and healthy donors were stained with anti-CD41 (HIP8; BioLegend, San Diego, CA, USA), anti-CD45 (2D1; eBioscience, San Diego, CA, USA), anti-PD-L1 (29E.2A3; BioLegend, San Diego, CA, USA), and Ghost Dye (Cell Signaling, Danvers, MA, USA). Prepared samples were run using CytoFlex LX (Beckman Coulter, San Diego, CA, USA) with the forward scatter threshold set at 0. Flow cytometry data were analyzed using the FlowJo™ version 10.7.1 Software (Tree Star, Ashland, OR, USA). Ghost Dye- CD45− CD41+ cells were considered to be platelets.

### 2.5. Statistical Analyses

The study endpoints were progression-free survival (PFS) and OS in patients in the DC and CCRT alone groups. PFS was defined as the time (in months) from the first date of CCRT (chemotherapy or RT) to the date of objective disease progression or death from any cause, irrespective of whether the patient was withdrawn from therapy or received another anticancer treatment before progression, and was censored at the date of the last patient contact. OS was measured from the first date of CCRT to the date of death for any reason and was censored at the date of the last patient contact.

Baseline characteristics were analyzed in the biomarker-evaluable population (BEP), which was defined as patients who received CCRT and for whom CTC and PBC analysis was available (Figure 1). Radiation pneumonitis and esophagitis were diagnosed clinically based on the presence of classic symptoms, timing and history of RT, imaging findings, and the exclusion of alternative causes [17]. All data are expressed as mean (standard deviation [SD]) for continuous variables and as numbers (percentages) for categorical variables. A comparative analysis of continuous and categorical variables between groups was performed using the Mann-Whitney U-test and Pearson’s chi-square or Fisher’s exact tests, respectively. The cut-off values for ANC, ALC, platelet, NLR, and PLR were calculated based on their median levels or analyses of receiver operating characteristic (ROC) curves and area under the curve (AUC). The optimal cutoff value was determined as the point at which the Youden index was at a maximum on the ROC curve. Survival times were estimated for each group using the Kaplan-Meier method. Univariable and multivariable survival analyses were performed using a Cox proportional hazards model. All statistical analyses were performed using SPSS ver. 25 (IBM Corp., Armonk, NY, USA) and MedCalc ver. 19.6 (MedCalc Software Ltd., Ostend, Belgium). A *p* < 0.05 was considered significant.

## 3. Results

### 3.1. Patient Enrollment and Baseline Characteristics

Among eighty-three patients diagnosed with stage III NSCLC between March 2020 and March 2021, we enrolled fifty patients with unresectable stage III NSCLC who started CCRT. A biomarker analysis using blood samples was performed in those populations (BEP) (Figure 1). Finally, 27 patients were assigned to the CCRT alone group, and 23 patients were assigned to the DC group.

The baseline characteristics and treatment-related factors of patients according to DC are summarized in Table 1. Squamous cell carcinoma was the predominant histologic type (60.0%). EGFR mutation and ALK translocation were detected in one and two patients, respectively. The results of PD-L1 immunohistochemistry (IHC) with SP263 antibody were confirmed in 49 patients. PD-L1 expression (TPS ≥ 1%) was found in 28 patients (56.0%), and most patients (95.7%) in the DC group showed PD-L1 TPS ≥1%. Most patients received five to six cycles of chemotherapy weekly with paclitaxel plus cisplatin (68.0%) or paclitaxel plus carboplatin (32.0%). The mean RT dose and fractions were 58.5 Gy and 29.0, respectively. The mean duration between the last RT and blood sampling at C1 was 32.5 days, and this time was longer in the CCRT alone group than in the DC group. Most patients (80.0%) had radiation pneumonitis, and steroid treatment was required in 10 patients (25.0%). Thyroiditis was the most common manifestation of immune-related adverse events. Nine patients completed the 12 month-course of DC without disease progression. Distant relapse occurred more frequently in the DC group than in the CCRT alone group, and platinum doublet chemotherapy was the most common modality as a subsequent treatment.

At the time of data cutoff (28 September 2022), the median follow-up duration was 27.4 months (95% CI: 26.7–28.1). The median PFS in the overall population was 13.1 months (95% CI: 8.6–17.5) with a maturity of 64.0% (32/50), and there was no significant difference in PFS between the CCRT alone group (median 13.5 months, 95% CI: 5.8–21.2) and the DC group (median 13.1 months, 95% CI: 6.9–19.2; *p* = 0.340) (Appendix A). The median OS in the overall population was 22.7 months with a maturity of 52.0% (26/50), and there was no significant difference in median OS between the CCRT alone group (21.0 months) and the DC group (25.1 months; *p* = 0.659) (Appendix A).

### 3.2. CTC as a Predictive Biomarker for DC

At the start of CCRT (C0), CTCs and CTC clusters per 7.5 mL of blood were detected in 34 (68.0%) and 18 patients (36.0%), respectively. The mean CTC and CTC cluster counts were 4.5 and 0.8, respectively (Appendix A). The CTC and CTC cluster counts increased after completion of CCRT (C0 to C1) and significantly decreased at the second follow-up CT scan (C1 to C5) (Figure 3a). In the serial monitoring of CTCs and CTC clusters, the DC group showed a constant range of fluctuations during treatment, while the CCRT alone group showed a steep decline from C1 to C5 (Figure 3b,c).

The median PFS and OS according to CTCs and CTC clusters counts are described in Appendix A. There was no difference in median PFS according to the presence of a CTC and CTC cluster at C0 (Appendix A). However, patients whose CTC and CTC clusters were detected at C1 showed numerically shorter median PFS than those with undetectable CTC and CTC clusters at C1 (Appendix A). Furthermore, patients with residual CTC clusters after CCRT at C1 showed worse PFS than those with cleared CTC clusters at C1, while there was no difference according to the clearance of CTC at C1 (Figure 4a,b). There was a significant difference in PFS according to the clearance of CTC clusters at C1 in the DC group, not in the CCRT alone group (Figure 4c,d). The OS did not vary according to the presence of CTC and CTC clusters at C0 (Appendix A).

### 3.3. PBC Count as a Potential Biomarker for DC

All parameters of PBC count decreased after completion of CCRT (C0 to C1), except PLR (Appendix A). ANC increased during DC or surveillance (C1 to C5), while other parameters of PBC did not show any significant difference in either group.

The median PFS and OS according to PBC counts are described in Appendix A. When divided by the median value of individual PBC parameters at C0 and C1, including ANC, ALC, platelet, NLR, and PLR, there was no difference in PFS between subgroups (Appendix A). However, based on the optimal cutoff value of platelet count analyzed by ROC curve analysis (Appendix A), patients with high platelets at C1 (PLT^hi^, >252 × 10^3^/µL) had a significantly worse median PFS than those with low platelets at C1 (PLT^lo^, ≤252 × 10^3^/µL) (Figure 4e). Unlike the results of CTC clusters, the PFS difference according to platelet count at C1 was consistently significant across both groups (Figure 4f,g), while the HR was higher in the CCRT alone group (4.88) than in the DC group (3.54) (Appendix A). Low ANC and low NLR at C0 and low platelets (≤252 × 10^3^/µL) at C1 were significantly associated with better OS (Appendix A).

### 3.4. Multivariable Analysis of PFS According to Blood-Based Biomarkers

Multivariable analyses revealed that residual CTC clusters at C1 (HR 3.49, 95% CI: 1.39–8.80) and PLT^hi^ at C1 (HR 5.59, 95% CI: 1.89–16.54) were significant risk factors for PFS. Survival analysis according to subgroups determined by these factors is described in Figure 5. In the DC group, the clearance of CTC clusters at C1 and platelet count at C1 was used to stratify patients into the following three subgroups: DC Group 1 (*n* = 8) with a cleared CTC cluster at C1 and low platelets at C1; DC Group 2 (*n* = 9) with a cleared CTC cluster at C1 and high platelets at C1, or residual CTC clusters at C1 and low platelets at C1; and DC group 3 (*n* = 2), with residual CTC clusters at C1 and high platelets at C1. In the CCRT alone group, patients were divided into two subgroups by platelet count at C1: CCRT group 1 (*n* = 20), with low platelets at C1, and CCRT group 2 (*n* = 6) with high platelets at C1. Kaplan-Meier analyses of PFS showed a clear difference among the five subgroups. Patients with DC who had cleared CTC clusters and PLT^lo^ at C1 (DC group 1) had the longest PFS, and those with DC who had residual CTC clusters and PLT^hi^ at C1 (DC group 3) showed the worst median PFS (2.6 months), worse than those without DC who had PLT^hi^ (CCRT group 2; 5.9 months).

### 3.5. PD-L1 Expression in Pbmcs and Relationship between Ctcs and Platelets

To understand the blood-based immune profiling of PBMCs and their association with PD-L1 expression in tumor cells, the proportions of PD-L1-expressed immune cells were analyzed by flow cytometry. When analyzing PBMC samples from NSCLC patients (C0) and healthy donors, NSCLC patients had up-regulated PD-L1 expression in platelets isolated from PBMCs, as shown on histograms compared with other circulating immune cells, and PD-L1 was not up-regulated in the platelets and immune cells in the PBMCs of healthy donors (Figure 6a). Furthermore, NSCLC patients had PD-L1-expressed-enriched platelets compared to healthy donors who had marginally expressed PD-L1 by platelets (Figure 6b). The proportions of PD-L1-expressed platelets at C0 were correlated with peripheral blood platelet count at C0 (r = 0.303, *p* < 0.05) (Appendix A). When divided into subgroups by the median value of platelet count at C0, the proportions of PD-L1-expressed platelets (PD-L1 + PLT %) at C0 were numerically higher in patients with high platelets than in those with low platelets at C0 (Appendix A).

Similar to CTCs, the circulating platelets were found to be EPCAM/CK-CD45-CD41+ cells using IF staining (Figure 6c). In the captured fluorescent images, CTCs also formed a cluster with CD41 + platelets as well as leukocytes, suggesting cell-to-cell interactions. Based on the optimal cut-off value of the proportions of PD-L1 + platelets (12.4%; AUC 0.677, *p* = 0.077) analyzed by the ROC curve, the proportions of patients with decreased CTC counts from C0 to C1 were significantly higher in the subgroup with low proportions of PD-L1 + platelets (≤12.4%) than in those with high proportions of PD-L1 + platelets at C0 (>12.4%) (Figure 6d).

## 4. Discussion

In this exploratory observational study, blood samples were prospectively collected from patients who started CCRT. We investigated the feasibility of a comprehensive analysis of CTC and PBC as predictive blood-based biomarkers for CCRT and DC in patients with unresectable stage III NSCLC. Overall, residual CTC clusters and PLT^hi^ at C1 were independent risk factors for PFS, and patients with DC who had residual CTC clusters and PLT^hi^ at C1 showed the worst median PFS, even worse than those without DC who had PLT^hi^. PD-L1 + platelets in PBMCs were more predominant in NSCLC patients than in healthy donors, and the low proportion of PD-L1 + platelets at baseline (C0) was associated with a reduction in CTC counts after CCRT (C1).

MRD detection reflects the presence of tumor cells in a patient after definitive treatment and a lack of any clinical or radiological signs of recurrence or metastasis [18]. The detection of MRD by using CTCs has been applied mainly to post-surgical early-stage lung cancers. Chemi et al. demonstrated that CTCs intraoperatively collected from a cancer-draining pulmonary vein (PV) were detected in 48% of patients enrolled in the TRACERx cohort [19], and the high PV-CTC (≥18) was an independent predictor of disease-free survival [10]. In a Taiwanese study of a surgery cohort, Wu et al. suggested that the early rebound of CTC counts on postoperative days 1 and 3 during longitudinal monitoring was associated with recurrence months later [9]. To the best of our knowledge, the present study is the first trial that attempts to characterize MRD using CTCs in unresectable locally advanced NSCLC. In the present study, most stage III NSCLC patients had CTCs (68.0%) even at diagnosis or before CCRT, suggesting that the reduction or clearance of CTCs after CCRT may be a predictive factor for disease progression or recurrence. The residual CTC clusters after CCRT was significantly associated with worse PFS in the overall population and DC group, while the differences in PFS according to the clearance of CTC clusters were not significant in the CCRT alone group. The survival and metastasis of CTCs have been reported to be associated with immune escape and interaction with circulating immune cells, such as T cells, NK cells, dendritic cells, macrophages, and myeloid-derived suppressor cells (MDSCs), and PBCs, such as neutrophils, red blood cells (RBCs), and platelets to survive in the blood microenvironment [20,21]. Therefore, the present study suggests that CTC may play a promising role in predicting the efficacy and surveillance of recurrence, especially during immunotherapy. However, the functional significance and predictive value of dynamic changes in CTCs in locally advanced NSCLC need to be validated in further studies.

The detection of MRD using ctDNA following CRT has been introduced previously in several studies. Chaundri et al. showed that cancer-personalized profiling by deep sequencing (CAPP-seq) analysis could detect ctDNA at a pre-specified MRD landmark (within four months after CRT) in 53% of patients, and the freedom from progression (FFP) at 36 months was 0% in patients with detectable and 93% in those with undetectable ctDNA MRD (HR 43.4; 95% CI: 5.7–341) [22]. Molding et al. demonstrated that patients with undetectable ctDNA MRD after CRT showed excellent FFP regardless of consolidation ICI, and patients with MRD after CRT receiving consolidation ICI had better FFP than those without consolidation ICI, supporting the personalization of consolidation ICI therapy [23]. ctDNA MRD analysis using genome-wide non-target NGS requires the development of a personalized or population-based library and a highly sensitive and specific detection technique. In comparison, as in the present study, CTC MRD analysis could confirm the MRD simply by identifying CTCs in whole blood at the landmark of post-CCRT, although there are still limitations in the standardization of the detection method and the inconvenience of long-term preservation. Therefore, as in our previous study [8], comprehensive analysis using blood components, such as CTC, PBC, and cfDNA, might be necessary to understand the landscape of the blood microenvironment, especially in the setting of ICI therapy. Furthermore, the application of MRD determined by these blood-based biomarkers to the clinical decision of patients with unresectable stage III NSCLC whether or not they receive DC after CCRT, needs to be validated by well-designed clinical trials.

CTC clusters consist of an aggregation of single CTCs (homotypic), or CTCs combined with neutrophils, MDSCs, tumor-associated macrophages, cancer-associated fibroblasts, other immune cells, and platelets (heterotypic) [24,25]. These non-malignant cells act as protectors and supporters for cancer cells in the blood and increase their ability to metastasize [25]. Compared to single CTCs, multiple pathways of cell-cell adhesion, stemness, and proliferation are up-regulated in CTC clusters [26]. In addition, CTC clusters have a more remarkable ability to metastasize, forming polyclonal metastatic foci [27], and exhibit stronger resistance to anti-tumor drugs than single CTCs [28]. The presence of CTC clusters has been known to be associated with worse clinical outcomes across the type of metastatic cancers [29]; however, there are few of these studies on lung cancer, especially locally advanced stage III NSCLC [30]. Recent clinical trials have been focused on CTCs and CTC clusters as therapeutic targets, especially in early and advanced breast cancer. These include targeting specific chemo-resistant CTCs expressing specific cytokeratin and cell adhesion molecules or reversing methylation patterns on essential stemness genes [24]. Continued studies on CTC clusters in lung cancer, regardless of the stage and histologic type, may be necessary to improve the overall prognosis.

As mentioned above, CTCs and platelets can form clusters that survive and migrate in the blood-immune microenvironment. In previous studies, it was shown that platelets protect tumor cells from lysis by different cytotoxic lymphocytes, including NK cells and effector T cells [31]. The activated platelets can physically shield CTCs by forming a layer of platelets from immune surveillance [32]. In addition, platelets can support tumor growth and metastasis through the secretion of various cytokines, such as transforming growth factor-beta (TGF-β) [33]. TGF-β inhibits the differentiation of T cells into cytotoxic T cells and up-regulates regulatory T cells (Tregs) [34], suggesting that high platelet counts may adversely affect the cellular immune system and immunotherapy in solid tumors. In several meta-analyses, pretreatment high platelet counts were a risk factor associated with unfavorable PFS and OS in lung cancer [35], and high PLR was significantly related to adverse ORR, PFS, and OS in patients with advanced NSCLC receiving immunotherapy [36]. In the present study, the formation of clusters with CTCs and platelets was seen on IF staining of PBMC samples (Figure 6c), and patients with PLT^hi^ before the start of DC (C1) were significantly associated with worse PFS compared to those with PLT^lo^.

A recent report suggested that PD-L1 can be expressed on human platelets, and platelet-derived PD-L1 was directly affected by anti-PD-L1 ICI (atezolizumab) [37]. In another study of a mouse model with PD-L1 knock-out tumor cells, platelet-derived PD-L1 could regulate and protect PD-L1-negative solid tumors from death by T cells and interference with platelet binding to PD-L1 negative cancer cells by anti-platelet agent (aspirin) promoted T cell-induced cancer cytotoxicity, assuming that a successful outcome of ICI in PD-L1 negative tumors may be explained by the presence of intra-tumoral platelets [38]. We assumed that PD-L1 expression on tumor-educated platelets might negatively affect the prognosis of patients. In the present study, peripheral platelet counts at baseline correlated with the proportion of PD-L1 + platelets in pretreatment PBMCs, and a lower proportion of PD-L1 + platelets before CCRT was associated with a reduction in CTC counts after CCRT. Although the efficacy of DC and the prognosis of patients with DC according to the proportion of PD-L1 + platelets could not be assessed in the present study, the presence of platelet-derived PD-L1 and PD-L1-expressed circulating platelets might be predictive biomarkers for ICI consolidation in locally advanced NSCLC. However, future preclinical studies are warranted to determine how platelets, which are anucleated cells, could express or induce PD-L1 protein expression, how platelet-derived PD-L1 plays a role in the immune escape of CTCs and CTC clusters, and whether PD-L1s expressed on circulating or intra-tumoral platelets differ in their roles.

This study has several limitations. First, unlike the PACIFIC trial [3], there was no difference in PFS or OS between the DC and CCRT alone groups. This may be due to a limited number of enrolled patients and not applying stratification or matching variables to enrollment, even though patients were prospectively recruited. Therefore, it is necessary to verify the candidate biomarkers of this study in a larger cohort. Second, the analysis of PD-L1 + platelets using blood samples obtained after the completion of CCRT was not performed. To prove the role of PD-L1 + platelets as a predictive biomarker to DC, comprehensive analyses using serial samples at the end of CCRT and during DC, which could reflect changes in the blood-immune microenvironment, are warranted in future studies. Third, we could not investigate PD-L1 expression on CTCs at the same time, since the cryopreservation of isolated CTCs and the immunofluorescence tagging for more than four surface markers were limited at the time of starting the present study due to the lack of corresponding technical facilities. There are a lot of reports indicating that PD-L1 expression in CTCs has prognostic value, and the comparison and relevance between CTCs and platelets in PD-L1 expression on their surface might need to be investigated. Finally, plasma cfDNA samples were processed and preserved during this study; however, the analysis of tumor burden or MRD using cfDNA or ctDNA was not performed before the data cutoff. The ctDNA MRD analysis needs to be supplemented to comprehensively analyze blood-based biomarkers for DC and other ICI therapy. They can be interpreted from the perspective of previous studies and of the working hypotheses. The findings and their implications should be discussed in the broadest context possible. Future research directions may also be highlighted.

## 5. Conclusions

This prospective exploratory study suggested the feasibility of CTCs and PBCs as predictive biomarkers in patients with unresectable stage III NSCLC who received CCRT and DC. The MRD after CCRT determined by the clearance of CTC clusters was associated with unfavorable PFS, and the impact was evident in the DC group rather than in the CCRT alone group. Patients with PLT^hi^ after CCRT showed significantly worse PFS and OS in both groups than those with PLT^lo^. A multivariable analysis showed that residual CTC clusters and PLT^hi^ after CCRT were independent risk factors for PFS, and patients with DC who had residual CTC clusters and PLT^hi^ after CCRT showed the worst median PFS, even worse than those without DC who had PLT^hi^. In addition, PD-L1 + platelets in PBMCs at baseline might be surrogates of MRD after CCRT. This comprehensive analysis of CTC and PBC counts could provide valuable insights into a biomarker-driven strategy for further management after the completion of CCRT, and could aid in the risk stratification of patients with unresectable stage III NSCLC who are eligible for DC during treatment. Future studies, particularly prospective studies with a larger cohort, are warranted to verify the clinical significance of these blood-based biomarkers in patients treated with CCRT and ICI.

## Figures and Tables

**Figure 1 cancers-15-01151-f001:**
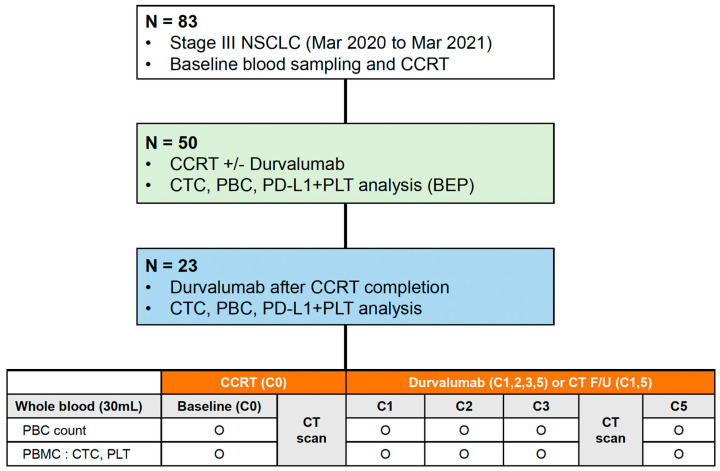
Study design and flow chart of the study. NSCLC, non-small cell lung cancer; CCRT, concurrent chemoradiotherapy; CTC, circulating tumor cell; PBC, peripheral blood cell; PD-L1, programmed death-ligand 1; PLT, platelet; BEP, biomarker-evaluable population; CT, computed tomography; PBMC, peripheral blood mononuclear cell.

**Figure 2 cancers-15-01151-f002:**
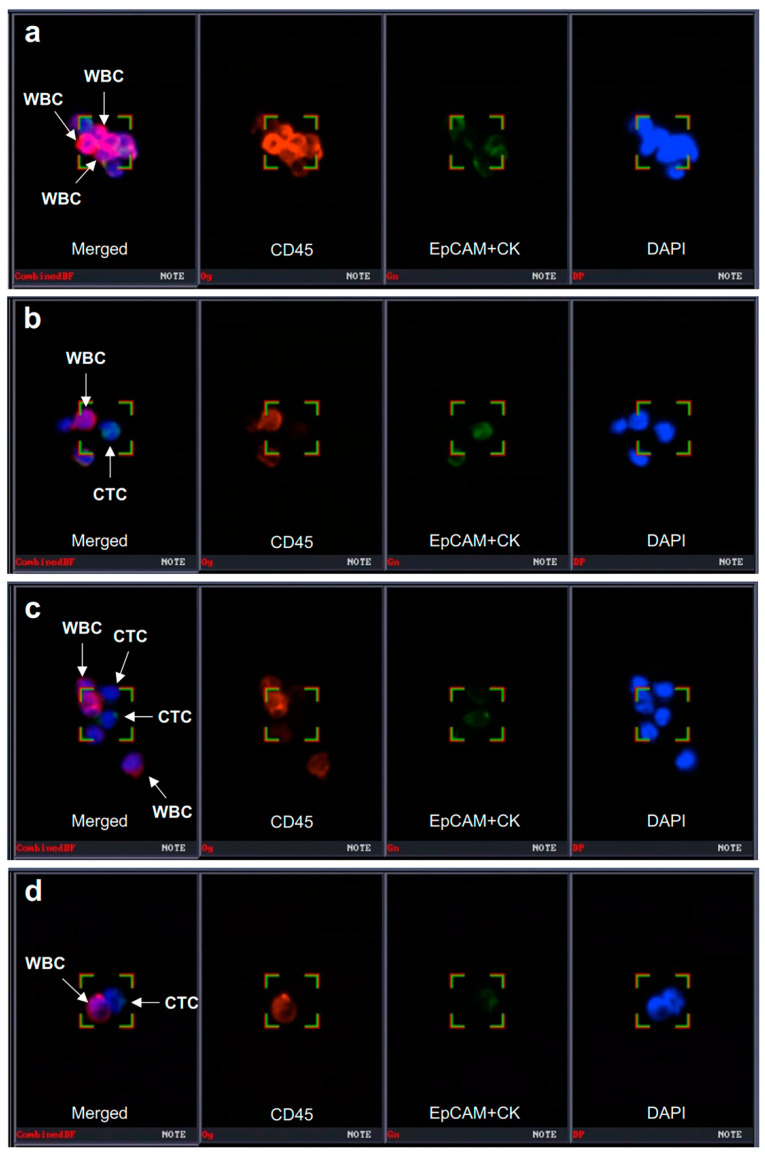
Identification of CTC and CTC clusters. CTC can be captured using a specific immunofluorescence staining system (BioViewCCBSTM). (**a**–**d**) CTC was identified as EpCAM/CK+ (epithelial cell surface markers) and CD45− cells (the first images from the left in each panel). CTCs can form a cluster with CD45+ leukocytes. CTC, circulating tumor cell; WBC, white blood cell; EpCAM, epithelial cell adhesion molecule; CK, cytokeratin; DAPI, 4′,6-diamidino-2-phenylindole. Magnification 40×.

**Figure 3 cancers-15-01151-f003:**
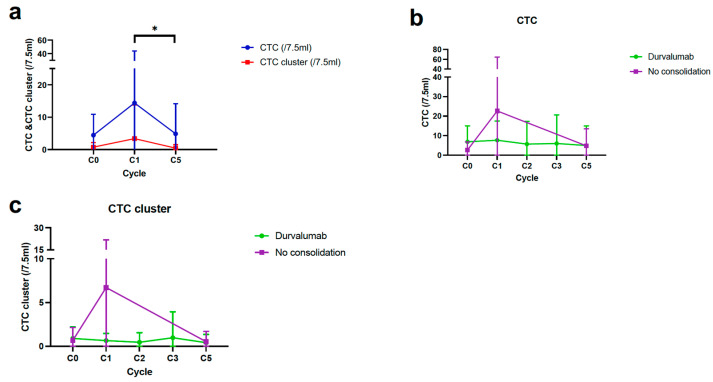
Longitudinal monitoring of CTCs and CTC clusters in patients with CCRT alone and durvalumab consolidation (DC). (**a**) Serial changes of CTCs and CTC clusters from C0 to C5 in overall patients. (**b**,**c**) Dynamic changes of CTCs and CTC clusters during follow-up according to DC. * *p* < 0.05. CTC, circulating tumor cell; CCRT, concurrent chemoradiotherapy.

**Figure 4 cancers-15-01151-f004:**
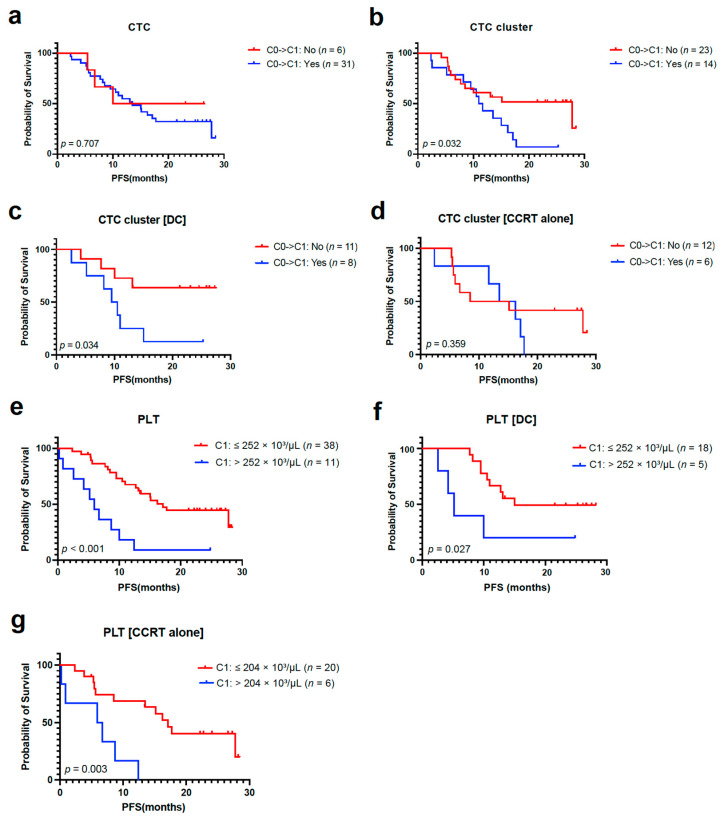
CTC, CTC cluster, and PBC count as predictive biomarkers for durvalumab consolidation (DC). (**a**,**b**) PFS analyses based on the clearance of CTCs and CTC clusters from C0 to C1 in the overall population. (**c**,**d**) PFS analysis according to clearance of CTC clusters from C0 to C1 in DC and CCRT alone group. (**e**–**g**) PFS analyses according to absolute PLT count at C1 in the overall population, DC, and CCRT alone group. CTC, circulating tumor cell; PBC, peripheral blood cell; PFS, progression-free survival; CCRT, concurrent chemoradiotherapy; PLT, platelet.

**Figure 5 cancers-15-01151-f005:**
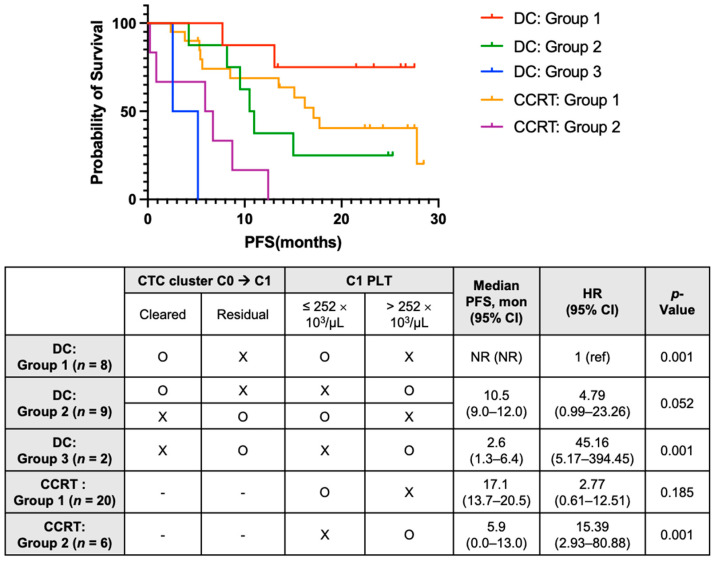
Multivariable analysis of PFS using blood-based biomarkers in patients treated with CCRT and durvalumab consolidation (DC) (in patients on DC). DC group 1 with cleared CTC cluster from C0 to C1 and low C1 PLT; DC group 2 with cleared CTC cluster from C0 to C1 and high C1 PLT, or residual CTC cluster at C1 and low C1 PLT; and DC group 3 with residual CTC cluster at C1 and high C1 PLT; (in patients on CCRT alone) CCRT group 1 with low C1 PLT and CCRT group 2 with high C1 PLT. PFS, progression-free survival; CCRT, concurrent chemoradiotherapy; CTC, circulating tumor cell; PLT, platelet; CI, confidence interval; HR, hazard ratio; NR, not reached; ref, reference.

**Figure 6 cancers-15-01151-f006:**
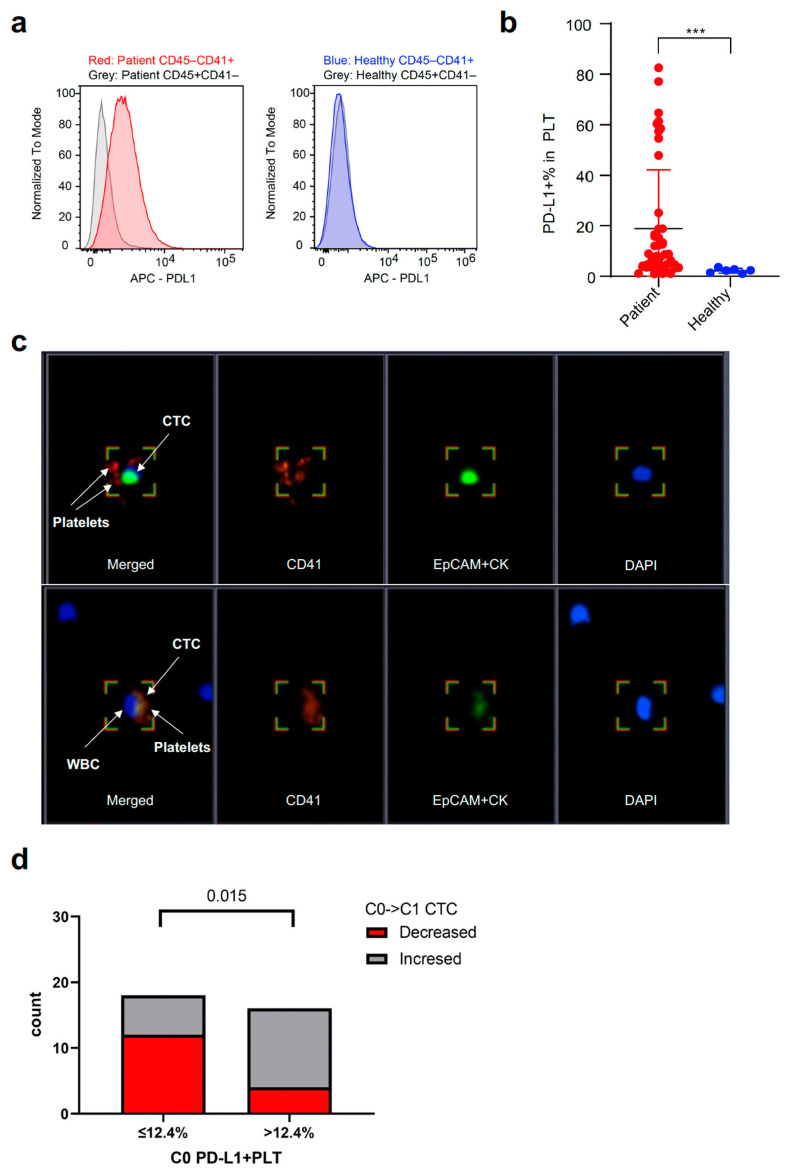
Relationship between CTCs and PD-L1 + PLT in PBMCs. (**a**,**b**) Difference in the expression of PD-L1 on PLT (included in PBMCs at C0) between NSCLC patients and healthy donors. CD45−CD41 + cells refer to PLT and CD45 + CD41−cells refer to immune cells. (**c**) PLTs can be captured and identified on immunofluorescence staining as EpCAM/CK−CD45−CD41 + cells. CTCs also form a cluster with CD41 + PLTs as well as leukocytes represented as DAPI + cells other than CTCs (lower right section). Magnification 40×. (**d**) Difference in the proportion of CTC changes from C0 to C1 according to PD-L1 + PLT (%) in PBMCs at C0. CTC, circulating tumor cell; PD-L1, programmed death-ligand 1; PLT, platelet; PMBC, peripheral blood mononuclear blood cell; NSCLC, non-small cell lung cancer; WBC, white blood cell; EpCAM, epithelial cell adhesion molecule; CK, cytokeratin; DAPI, 4′,6-diamidino-2-phenylindole. *** *p* < 0.001.

**Table 1 cancers-15-01151-t001:** Baseline characteristics of patients on concurrent chemoradiotherapy (CCRT) alone and with durvalumab consolidation (DC).

Characteristics	Total (*n* = 50)	CCRT Alone (*n* = 27)	DC (*n* = 23)	*p*-Value
Age	68.2 (7.9)	68.5 (7.7)	67.2 (8.5)	0.740
Sex				0.322
Female	4 (8.0)	1 (3.7)	3 (13.0)	
Male	46 (92.0)	26 (96.3)	20 (87.0)	
Smoking				0.900
Never smoker	3 (6.0)	2 (7.4)	1 (4.3)	
Current smoker	28 (56.0)	15 (55.6)	13 (56.5)	
Ex-smoker	19 (38.0)	10 (37.0)	9 (39.1)	
Pack × Years	39.8 (19.8)	37.1 (17.3)	43.4 (22.4)	0.277
Histology				0.402
ADC	16 (32.0)	10 (37.0)	6 (26.1)	
SQC	30 (60.0)	14 (51.9)	16 (69.6)	
NSCLC, NOS	4 (8.0)	3 (11.1)	1 (4.3)	
Stage (TNM 8th)				0.906
IIIA	23 (46.0)	12 (44.4)	11 (47.8)	
IIIB	19 (38.0)	11 (40.7)	8 (34.8)	
IIIC	8 (16.0)	4 (14.8)	4 (17.4)	
EGFR mutation				0.264
Wild type	22 (44.0)	14 (51.9)	8 (34.8)	
Mutant	1 (2.0)	1 ^1^ (3.7)	0 (0.0)	
Unknown	27 (54.0)	12 (44.4)	15 (65.2)	
ALK translocation				0.442
Negative	20 (40.0)	13 (48.1)	7 (30.4)	
Positive	2 (4.0)	1 (3.7)	1 (4.3)	
Unknown	28 (56.0)	13 (48.1)	15 (65.2)	
PD-L1 IHC (SP263)				<0.001
TPS < 1%	21 (42.0)	20 (74.1)	1 (4.3)	
TPS ≥ 1%	28 (56.0)	6 (22.2)	22 (95.7)	
Unknown	1 (2.0)	1 (3.7)	0 (0.0)	
Chemotherapy Rx				0.827
Pac-Cis	34 (68.0)	18 (66.7)	16 (69.6)	
Pac-Car	16 (32.0)	9 (33.3)	7 (30.4)	
Chemotherapy cycle	5.4 (1.3)	5.1 (1.6)	5.9 (0.4)	0.035
RT fraction	29.0 (4.1)	28.2 (5.8)	30.0 (0.0)	0.032
RT dose, Gy	58.5 (8.4)	57.5 (12.2)	59.8 (0.7)	0.711
Last RT to C1 ^2^ sample	32.5 (16.9)	39.5 (20.4)	27.1 (7.9)	0.004
CCRT best response				0.221
CR	0 (0.0)	0 (0.0)	0 (0.0)	
PR	33 (66.0)	16 (59.3)	17 (73.9)	
SD	14 (28.0)	8 (29.6)	6 (26.1)	
PD	3 (6.0)	3 (11.1)	0 (0.0)	
ORR, %	66.0	59.3	73.9	
Radiation pneumonitis	40 (80.0)	21 (77.8)	19 (82.6)	0.736
On steroid treatment	10 (25.0)	4 (19.0)	6 (31.6)	0.473
Radiation esophagitis	24 (48.0)	13 (48.1)	11 (47.8)	0.982
IrAEs	-	-	13 (56.5)	-
Thyroiditis	-	-	9 (39.1)	-
Skin eruption	-	-	2 (8.7)	-
Fever	-	-	1 (4.3)	-
Pericardial effusion	-	-	1 (4.3)	-
Pneumonitis	-	-	1 (4.3)	-
Hearing loss	-	-	1 (4.3)	-
Progression	32 (64.0)	19 (70.4)	13 (56.5)	0.309
PD during CCRT	-	2 (7.4)	-	-
DC completion	-	-	9 (39.1)	-
PD during DC	-	-	11 (47.8)	-
Progression type				0.169
Localized/Regional	17 (53.1)	12 (63.2)	5 (38.5)	
Distant	15 (46.9)	7 (36.8)	8 (61.5)	
Post-PD treatment (1st)				0.398
OP or RT	2 (4.0)	1 (3.7)	1 (4.3)	
Chemotherapy ^3^	16 (32.0)	7 (25.9)	9 (39.1)	
ICIs	3 (6.0)	3 (11.1)	0 (0.0)	
TKIs	3 (6.0)	2 (7.4)	1 (4.3)	
BSC or loss of follow-up	8 (16.0)	6 (22.2)	2 (8.7)	

Values are presented as mean (SD) or number (%). ^1^ L858R. ^2^ the first follow-up CT scan after CCRT in CCRT alone group and the first cycle of durvalumab in DC group. ^3^ platinum-doublet. SD, standard deviation; ADC, adenocarcinoma; SQC, squamous cell carcinoma; EGFR, epidermal growth factor receptor; ALK, anaplastic lymphoma kinase; PD-L1, programmed cell death ligand-1; IHC, immunohistochemistry; TPS, tumor proportional score; Rx, regimen; Pac-Cis, paclitaxel plus cisplatin; Pac-Car, paclitaxel plus carboplatin; RT, radiotherapy; CR, complete response; PR, partial response; SD, stable disease; PD, progressive disease; ORR, objective response rate; IrAE, immune-related adverse event; OP, operation; ICI, immune checkpoint inhibitor; TKI, tyrosine kinase inhibitor; BSC, best supportive care.

## Data Availability

The data presented in this study are available on request from the corresponding author. The data are not publicly available due to institutional data-sharing restrictions.

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
