# Peer review of "Blood-Based Biomarker Analysis for Predicting Efficacy of Chemoradiotherapy and Durvalumab in Patients with Unresectable Stage III Non-Small Cell Lung Cancer"

_cancers, 2023, doi:10.3390/cancers15041151_

Round 1

Reviewer 1 Report

The manuscript entitled “Blood-Based Biomarker Analysis for Predicting Efficacy of 2 Chemoradiotherapy and Durvalumab in Patients with Unresec-3 table Stage III Non-Small Cell Lung Cancer” is interesting. The authors have shown that CTCs clusters and the number of PLTs have prognostic value for NSCLC patients with unresectable tumors who received CCRT (with or without DC treatment), however, there are some issues that need to be addressed.

In the abstract, the authors must include the p-values for PFS etc.

Why did the authors decide to check the PD-L1 expression in PLTs and not in CTCs? There are a lot of manuscripts describing that PD-L1 expression in CTCs also has prognostic value. This should be discussed in the corresponding section.

Is there a significant correlation between the CTCs with PLTs expressing PD-L1?

In Figure 2 the images are rather low, and the letters are very small. Furthermore, it is necessary to add numbers or letters to be easier to follow the legend.

In figure 5e the photos are unclear and the magnification is low.

The information in Figures 5c and 5d is unimportant and could be moved to suppl. Figures.

On contrary, it would be interesting to show the changes in CTCs numbers or CTCs clusters in a histogram.

Author Response

Manuscript ID: cancers-2193036

Title: Blood-based Biomarker Analysis for Predicting Efficacy of Chemoradiotherapy and Durvalumab in Patients with Unresectable Stage III Non-Small Cell Lung Cancer

Dear the Editors of Cancers,

Thank you for giving us the opportunity to submit a revised manuscript again.
We appreciate the time and effort that you and the reviewers dedicated to providing feedback on our manuscript and are grateful for the insightful comments to improve our paper. We have incorporated an additional suggestion made by the reviewers.

The revised manuscript was written used by ‘Track changes’ and we reply as point-by-point response to the reviewer’s comment. You can enable the "Simple Markup" function from the "Track Changes" section of the "Review" tab. With simple markup enabled, all edits appear as red vertical lines, which you can click on to reveal individual changes.

We hope that all changes we have made meet with your approval and look forward to your response.

Sincerely yours,

In-Jae Oh, on behalf of all the authors.

Department of Internal Medicine, Chonnam National University Hwasun Hospital, 322 Seoyang-ro, Hwasun, Jeonnam 58128, Republic of Korea

Tel.: 061-379-7617, Fax: 061-379-7619, E-mail: droij@jnu.ac.kr

[Reviewer 1`s comments]
The manuscript entitled “Blood-Based Biomarker Analysis for Predicting Efficacy of Chemoradiotherapy and Durvalumab in Patients with Unresectable Stage III Non-Small Cell Lung Cancer” is interesting. The authors have shown that CTCs clusters and the number of PLTs have prognostic value for NSCLC patients with unresectable tumors who received CCRT (with or without DC treatment), however, there are some issues that need to be addressed.

Comment 1: In the abstract, the authors must include the p-values for PFS etc.

Reply 1: We described p-values for PFS and multivariable analysis.

Changes in the abstract: At median follow-up of 27.4 months, patients with residual CTC clusters at C1 had worse median PFS than those without detectable CTC cluster (11.0 vs. 27.8 months, p=0.032), and this trend was noted only in DC group (p=0.034). Patients with high platelets at C1 (PLThi, >252â…¹103/µL) had worse median PFS than those with low platelets (PLTlo) (5.9 vs. 17.1 months, p<0.001). In multivariable analysis, PLThi and residual CTC clusters at C1 were independent risk factors for PFS, and DC group with PLThi and residual CTC clusters at C1 showed the worst median PFS (2.6 months, HR 45.16, p=0.001), even worse than that of CCRT alone group with PLThi (5.9 months, HR 15.39, p=0.001).

Comment 2: Why did the authors decide to check the PD-L1 expression in PLTs and not in CTCs? There are a lot of manuscripts describing that PD-L1 expression in CTCs also has prognostic value. This should be discussed in the corresponding section.

Reply 2: Thank you for your valuable comment. When we started the present study, the cryopreservation of isolated CTCs and the immunofluorescence tagging for more than four surface markers were limited due to the lack of corresponding technical facilities. Thus, we could not investigate PD-L1 expression on CTCs at the same time as our analysis. Meanwhile, the peripheral blood platelets and CTC clusters were demonstrated as predictive biomarkers in the present study, and we decided to investigate the association between those two biomarkers. We assumed that PD-L1 expression on tumor-educated platelets might negatively affect the clearance of CTC clusters after CCRT and the prognosis of patients, and our study showed a possibility of PD-L1+platelets as a potential biomarker in patients with CCRT and durvalumab consolidation. We described the aforementioned explanations in the discussion section.     

Changes in the text: (Discussion, page 13, line 421-423) We assumed that PD-L1 expression on tumor-educated platelets might negatively affect the prognosis of patients.

(Discussion, page 13, line 426-429) Although the efficacy of DC and the prognosis of patients with DC according to the proportion of PD-L1+platelets could not be assessed in the present study, the presence of platelet-derived PD-L1 and PD-L1-expressed circulating platelets might be predictive biomarkers for ICI consolidation in locally advanced NSCLC.

(Discussion, page 14, line 443-448) Third, we could not investigate PD-L1 expression on CTCs at the same time, since the cryopreservation of isolated CTCs and the immunofluorescence tagging for more than four surface markers were limited at the time of starting the present study due to the lack of corresponding technical facilities. There are a lot of reports describing that PD-L1 expression in CTCs has prognostic value, and the comparison and relevance between CTCs and platelets in PD-L1 expression on their surface might need to be investigated.

Comment 3: Is there a significant correlation between the CTCs with PLTs expressing PD-L1?

Reply 3: We investigated the correlation between CTC counts and the proportions of PD-L1+platelets, and the results are described in the Figure 6 (rearranged) and the corresponding manuscript (page 10, line 308-315). Based on the cut-off value of the proportions of PD-L1+platelets by ROC curve, the proportions of patients with decreased CTC counts from C0 to C1 were significantly higher in the subgroup with a low % of PD-L1+platelets than in those with a high % of PD-L1+platelets at C0. In fact, the analysis of PD-L1+platelets using serial samples after the completion of CCRT might be needed to confirm the relevance between PD-L1+platelets and the efficacy of CCRT or durvalumab consolidation. However, this finding might be valuable in that PD-L1+platelets in PBMCs at baseline might be surrogates of minimal residual disease after CCRT, and further studies using on-treatment samples are warranted.    

Comment 4: In Figure 2 the images are rather low, and the letters are very small. Furthermore, it is necessary to add numbers or letters to be easier to follow the legend.

Reply 4: We magnified the figures and added letters to each panel. We also modified the figure legend accordingly. In addition, we submitted a revised file for Figure 2.

Changes in the text: (Results, Page 5, line 176-177) (a-d) CTC was identified as EpCAM/CK+ (epithelial cell surface markers) and CD45− cells (the first images from the left in each panel).

Comment 5: In figure 5e the photos are unclear and the magnification is low.

Reply 5: We selected two clear images in accordance with the results described in the manuscript. We magnified them and also modified the figure legend accordingly. In addition, we submitted a revised file as Figure 6c (rearranged).

Changes in the text: (Results, page 11)

Comment 6: The information in Figures 5c and 5d is unimportant and could be moved to suppl. Figures.

Reply 6: We removed Figure 5c and 5d and rearranged to Supplementary Figure S4a and S4b. We modified the orders of figures and figure legend accordingly. In addition, we submitted a revised file.

Comment 7: On contrary, it would be interesting to show the changes in CTCs numbers or CTCs clusters in a histogram.

Reply 7: Thank you for your comment. We changed Supplementary Figure S2 to Figure 3 (rearranged), which showed longitudinal monitoring of dynamic changes in the numbers of CTCs and CTC clusters. Accordingly, we added the corresponding legend of the rearranged Figure 3 and modified the order of figure numbers to the flow of the manuscript. In addition, we submitted a revised file.

Reviewer 2 Report

This is a well-designed and written manuscript titled “Blood-Based Biomarker Analysis for Predicting Efficacy of 2 Chemoradiotherapy and Durvalumab in Patients with Unresectable Stage III Non-Small Cell Lung Cancer”. The author reports in this manuscript about the feasibility of circulating tumor cells (CTCs) and peripheral blood cells (PBCs) count, as predictive biomarkers in patients with unresectable stage III non-small cell lung cancer (NSCLC) who received concurrent chemoradiotherapy (CCRT) and durvalumab. Comprehensive analysis of CTCs and PBCs counts before and after CCRT provides valuable insights and can serve as a potential biomarker for predicting survival and management. This manuscript will be very useful for the readers and can be accepted in present form. 

Author Response

Manuscript ID: cancers-2193036

Title: Blood-based Biomarker Analysis for Predicting Efficacy of Chemoradiotherapy and Durvalumab in Patients with Unresectable Stage III Non-Small Cell Lung Cancer

Dear the Editors of Cancers,

Thank you for giving us the opportunity to submit a revised manuscript again.
We appreciate the time and effort that you and the reviewers dedicated to providing feedback on our manuscript and are grateful for the insightful comments to improve our paper.

Sincerely yours,

In-Jae Oh, on behalf of all the authors.

Department of Internal Medicine, Chonnam National University Hwasun Hospital, 322 Seoyang-ro, Hwasun, Jeonnam 58128, Republic of Korea

Tel.: 061-379-7617, Fax: 061-379-7619, E-mail: droij@jnu.ac.kr

Round 2

Reviewer 1 Report

The authors responded adequately to my comments